# Rethinking Fair Federated Learning from Parameter and Client View

**Kaiqi Guan**[1][†], **Wenke Huang**[1][†], **Xianda Guo**[1],
**Yueyang Yuan**[1], **Bin Yang**[1], **Mang Ye**[1][*]

[1] National Engineering Research Center for Multimedia Software, Institute of Artificial Intelligence,
Hubei Key Laboratory of Multimedia and Network Communication Engineering,
School of Computer Science, Wuhan University, Wuhan, China.
`{guankaiqi, wenkehuang, yemang}@whu.edu.cn`

## Abstract

Federated Learning is a promising technique that enables collaborative machine learning while preserving participant privacy. With respect to multi-party collaboration, achieving performance fairness acts as a critical challenge in federated systems. Existing explorations mainly focus on considering all parameter-wise fairness and consistently protecting weak clients to achieve performance fairness in federation. However, these approaches neglect two critical issues. 1) Parameter Redundancy: Redundant parameters that are unnecessary for fairness training may conflict with critical parameters update, thereby leading to performance degradation. 2) Persistent Protection: Current fairness mechanisms persistently enhance weak clients throughout the entire training cycle, hindering global optimization and causing lower performance alongside unfairness. To address these, we propose a strategy with two key components: First, parameter adjustment with mask and rescale which discarding redundant parameter and highlight critical ones, preserving key parameter updates and decrease conflict. Second, we observe that the federated training process exhibits distinct characteristics across different phases. We propose a dynamic aggregation strategy that adaptively weights clients based on local update directions and performance variations. Empirical results on single-domain and cross-domain scenarios demonstrate the effectiveness of the proposed solution and the efficiency of crucial modules. The code is available at `https://github.com/guankaiqi/FedPW`.

## 1 Introduction

Federated Learning (FL) is a collaborative machine learning framework [26, 57, 30, 27, 58, 19] that enables multiple clients to jointly train a global model [38, 31, 18] without sharing raw data. Clients process data locally and periodically send model updates to the server, which aggregates these updates into a global model. This training paradigm effectively addresses data island and privacy issues. However, due to data heterogeneity [58, 23], intermittent client participation, and system heterogeneity, the model is prone to unfairness, which diminishes FL's generalization capability.

Improving **performance fairness** [47, 19] is a central research focus in federated learning. Existing approaches can be categorized into three types: client selection [16, 42], weight allocation [39, 35, 48], and personalized local models [33]. For instance, to enhance the performance of underperforming clients, the federated server may assign them larger aggregation weights, amplifying their influence

---

[†] Equal Contribution.
[*] Corresponding Author.

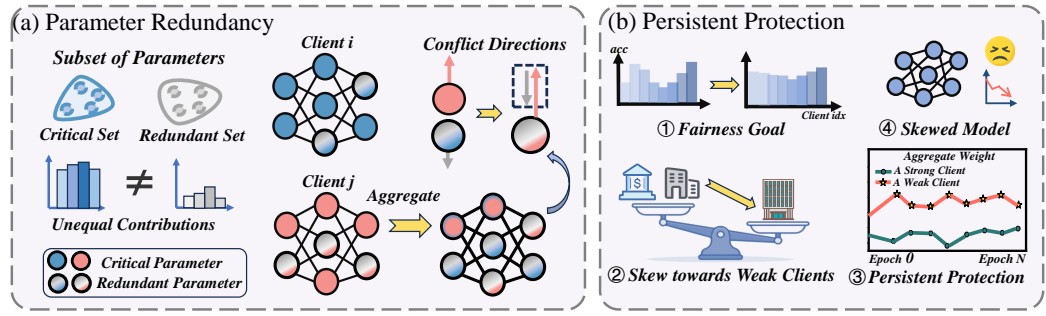

Figure 1: **Problem illustration** of existing fairness methods: Traditional approaches consider all model parameters for fairness, but not all parameters are equally important. Redundant parameters can conflict with key ones, disrupting their crucial contributions. Additionally, a long-term weighting strategy that favors weak clients is a classic approach. However, since weak clients often deviate from the global trend, this strategy can lead to a skewed initial training direction, resulting in an undertrained model and performance degradation.

on model updates. However, these methods typically compromise the performance of the global model. But training an effective global model is the primary goal of FL. This provokes our thinking:

> *Can we design an algorithm for FL that promotes fairness while improving the performance of the global model?*

We identify two primary causes of global model degradation: **I Parameter Redundancy**: *Due to the high redundancy in model parameters, a significant portion of parameters are inherently non-essential, making it unnecessary for all parameters to participate in model aggregation.* Owing to the over-parameterized characteristics of deep neural networks [10, 51, 59], not all parameters contribute equally to fitting the domain distribution. Specifically, only a subset of critical core parameters plays a decisive role in model training, while other non-core, less important parameters introduce noise that disrupts the parameter space. Moreover, these marginal parameters are prone to conflicting with other critical parameters, leading to performance degradation and unfairness. **II Persistent Protection**: *Fairness approaches typically protect weak clients throughout the entire training cycle, leading to suboptimal model optimization.* From a conventional fairness perspective, the existence of weak clients is paramount to ensuring equitable performance across heterogeneous participants. However, we argue that such clients may inherently act as outliers within the federated ecosystem. If the system rigidly applies a static prioritization strategy favoring underperforming clients across all training phases, the global model may become excessively influenced by their gradient directions during early stages, leading to a deviation in the model's optimization trajectory and resulting in performance degradation and unfairness.

To address the aforementioned problems, we propose FedPW( Fair **Fed**erated Learning via **P**arameter Adjustment and Adaptive **W**eighting). For problem **I**, we leverage Parameter Adjustment to address this challenge. Specifically, we observe that discarding small updates to parameters can reduce conflicts with clients from other domains without negatively affecting the model's performance on its own domain. Interestingly, we also find that different domains exhibit varying levels of tolerance for parameter discarding, which is inversely correlated with the domain's complexity. Therefore, we apply a domain-specific drop rate to discard the tail parameters of each client, ensuring fairness in the discarding process while mitigating conflicts between domains. However, the effect of the discarding operation is limited. To further reduce confusion during aggregation, we identify a set of consensus parameters for amplification, making the global model's update direction more stable and consistent, while scaling the discarded parameter updates back to their original magnitude, which is shown to benefit the model's performance [60, 15].

For Problem **II**, we propose an adaptive weighting mechanism that dynamically adjusts model aggregation weights based on the evolving training dynamics. We observe that the training process exhibits distinct phases. Early stages may benefit from reinforcing consensus to stabilize joint training, while later stages require emphasis on domain diversity to enhance fairness. To implement this insight, we dynamically allocate aggregation weights using the dot product between client parameter updates and loss variations. Our method achieves a transition from reinforcing consensus to emphasizing fairness, resulting in excellent performance and fairness. The details are presented in Sec. 3.3.

In this paper, FedPW consists of two main components. First, parameter adjustment alleviates parameter update conflicts by discarding certain parameters, and strengthens consensus to emphasize important parameters across multiple parties. Second, through the reweighting strategy, FedPW ensures balanced performance across clients from different domains, guaranteeing fairness in the diverse update directions during multi-party collaboration. The method is straightforward to implement and focuses on improving the aggregation step, making it easily compatible with other federated learning methods. The main contributions are summarized as follows:

❶ *Re-examining Why might Fairness Methods Harm Model Performance*. Full parameter participation introduce unnecessary conflicts via redundant parameters, undermining model training.

❷ *A Novel Partial-parameter Dynamic Aggregation Framework*. We discard minor updates while amplifying critical ones to adjust the model parameters. Furthermore, we propose an adaptive weighting strategy based on the dynamic characteristics of the training process, which mitigates the negative impact of lagging clients hindering the global model during the early stages of training.

❸ *Extensive Experimental Validation*. We conduct experiments on single-domain and cross-domain scenarios. With ablations, we validate the efficacy of FedPW and the indispensability of modules.

## 2 Related Work

### 2.1 Heterogeneous Federated Learning

Statistical heterogeneity across parties, commonly referred to as the non-IID problem, poses significant challenges in Federated Learning (FL). The pioneering work FedAvg [38] demonstrated notable performance degradation under heterogeneous data settings. To address this, many approaches employ regularization terms to constrain local training. For instance, FedProx [34] introduced a proximal term to mitigate divergence between local and global models, while FedDyn [1], FedCurv, and pFedMe [49] adopted similar regularization strategies. Methods like MOON [31], FCCL [17], FedUFO [62], FedProto [50], FPL [18], and FedProc [40] incorporate alignment-based penalty terms to harmonize feature representations across clients, addressing data heterogeneity. SCAFFOLD [24] proposed a control variate mechanism to correct client drift by reducing gradient divergence. Other approaches tackle heterogeneity via prototype-based communication. FedProto [50] aligns global and local class prototypes to handle label distribution skew, though it primarily targets single-domain label skew. For cross-domain challenges, FPL [18] leverages clustered prototypes to generate unbiased global representations, while FedGA [63] and FedDG [36] focus on domain generalization for unseen target domains. However, these methods often involve full parameter updates during training, which introduces redundancy. Our approach emphasizes parameter adjustment to prioritize critical parameters, effectively resolving conflicts arising from redundant parameters across diverse domains.

### 2.2 Fair Federated Learning

Fairness has been a key focus in Federated Learning, with various concepts proposed, such as Performance Fairness [39, 22], Collaboration Fairness [37, 64, 55], and Group Fairness [9, 61, 6]. Performance Fairness, which aims to ensure similar accuracy across clients, is one of the most widely studied areas [32, 39]. Some methods address this by modifying client selection strategies. For instance, UCB-CS [5] uses a communication-efficient selection strategy based on multi-armed bandit theory, choosing clients with higher local loss to promote fairness and consistency. Other approaches focus on adjusting aggregation weights to unify training outcomes. A notable example is AFL [39], which minimizes maximum loss to improve the performance of the worst-performing devices. In contrast, q-FFL [35] introduces exponentially scaled weights to penalize clients with higher loss, leading to a more balanced accuracy distribution. FedHEAL [3] leverages the distance between local models and the global model to constrain unfair disparities. FedFV [53] uses cosine similarity to detect and resolve gradient conflicts iteratively, converging to a Pareto-stable solution. Ditto proposes a personalized federated learning framework that employs a penalty term to control the degree of model personalization, thereby achieving fairness and robustness. FedCE [22] leverages client contribution estimation as global model aggregation weights, demonstrating improved Performance Fairness and Collaboration Fairness. However, these methods often exhibit a persistent tendency to protect underperforming clients throughout the entire training cycle, which may hinder model training. Our method leverages the training dynamics, applying different weighting strategies at different stages, thereby achieving dual excellence in both generalization and fairness.

# 3 Methodology

## 3.1 Preliminary

**Federated Learning and Performance Fairness**. In typical federated learning, a system consists of $K$ clients, each with private data $D_k = \{x_i, y_i\}_{i=1}^{N_k}$. At the start of each communication round $t$, the server shares the global model $w^t$ with all clients. Each client initializes its local model $w_k^t$ with $w^t$, performs local optimization using its data, and sends the updated parameters back to the server. The server then aggregates these updates using weighted averaging:

$$w_k^t \leftarrow w_k^t - \eta \nabla \sum_{i \in B_k} l(w_k^t, \xi_i), \quad w^{t+1} = \sum_k \lambda_k w_k^t. \tag{1}$$

Here, $B_k$ is a mini-batch sampled from the local dataset $D_k$, $\xi$ represents a query instance, and $\eta$ is the local learning rate. The optimization objective is to minimize global loss:

$$\min_w F(w) = \sum_{k=1}^{K} \lambda_k f_k(w), \tag{2}$$

where $\lambda_k$ is the weight of client and $f_k(w)$ is the loss of local model with parameters $w$.

**Definition 3.1. (Performance Fairness)** *Given two trained models, $w$ and $\tilde{w}$, model $w$ is considered to provide a fairer solution to the federated learning objective (2) if its performance across the $m$ devices is more uniform compared to model $\tilde{w}$, i.e.* $\text{var} \{F_k(w)\}_{k \in [K]} < \text{var} \{F_k(\tilde{w})\}_{k \in [K]}$.

## 3.2 Parameter Adjustment

**Motivation.** Previous work has shown that model parameters often contain significant redundancy [60, 15, 2, 7, 8, 52], with only a small set of key parameters driving performance, while most parameters are redundant and ineffective. To investigate parameter redundancy, we conducted an observational experiment on the Digits dataset. We trained on 5 clients across four domains for 20 epochs,

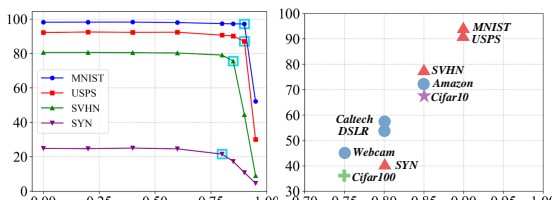

Figure 2: **Study on Parameter Redundancy**. Left: Model accuracy declines with increasing mask ratio. Right: Redundancy-performance relationship across datasets.

gradually increasing the mask rate while monitoring performance. As shown in Fig. 2, all domains showed almost no performance change when a small number of parameters were discarded, with performance degradation only occurring when a large proportion of parameters was discarded, indicating the redundancy of the parameters. Our findings revealed that each domain exhibited significant parameter redundancy, and domains with lower performance had less redundancy(Fig. 2). If redundant parameters are included in aggregation, they can overwhelm important updates, leading to confusion in the global model. Therefore, minimizing this redundancy is essential to ensure effective model updates. Our approach focuses on discarding unimportant parameters and enhancing the updates of key ones. The specific process is as follows:

**Selection of Unimportant Parameters**. As previously established, the parameter updates $\Delta w_k^t$ exhibit significant variation: while most parameters undergo negligible changes ($|\Delta w_{k,i}^t| \rightarrow 0$), a critical minority demonstrate substantial updates. We prune insignificant parameters by first representing the $G$-dimensional update vector:

$$\Delta w_k^t = [\Delta w_{k,1}^t, \ldots, \Delta w_{k,G}^t]. \tag{3}$$

Unimportant parameters are defined as those below threshold $\tau_k = \text{sorted}(|\Delta w_k^t|)[(1 - r_k^t)G]$, where $r_k^t \in (0, 1]$ is the client-specific mask rate. As demonstrated in Fig. 2, parameter redundancy inversely correlates with client performance (quantified by training loss). In our experiments, various methods to increase the parameter redundancy for clients with lower loss were proven effective, with the inverse of loss being the simplest and most effective approach. Thus, we compute $r_k^t$ using the inverse of smoothed loss $q_k^t$ from Eq. (9), where $c$ is a hyper-parameter representing the average mask ratio.

$$r_k^t = c \cdot \frac{1/q_k^t}{\sum_{k=1}^{K} 1/q_k^t}. \tag{4}$$

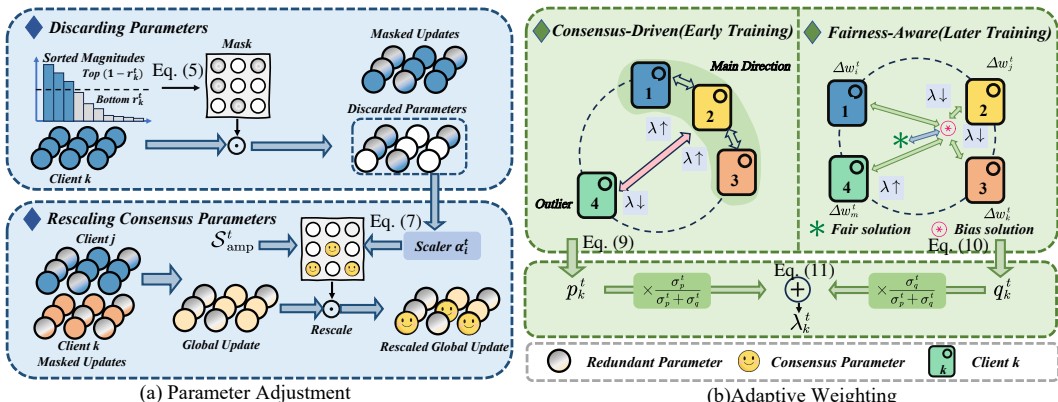

Figure 3: **Architecture illustration** of FedPW. FedPW consists of two core components: ❶ The left box refers to **Parameter Adjustment (PA)**, which adaptively discards redundant parameters and amplifies consensus ones (Sec. 3.2), reducing interference among parameters. ❷ The right box represents **Adaptive Weighted Aggregation (AWA)**, where we dynamically assign weights to each client according to the evolving training process (Sec. 3.3). In this way, conflicts among clients can be mitigated while reinforcing consensus.

**Discarding Redundant Parameters**. We use a mask $\mathcal{M}_{k,i}^t \in \mathbb{R}^G$ based on magnitudes to focus on important parameters which is used to discard the smallest subset of updates. The mask is defined as:

$$\mathcal{M}_{k,i}^t = \begin{cases} 1, & \text{if } |\Delta w_{k,i}^t| \geq \tau_k, \\ 0, & \text{otherwise.} \end{cases} \tag{5}$$

Finally, the masked updates are $\Delta \mathbf{w}_k^t = \Delta w_k^t \odot \mathcal{M}_k$, where $\odot$ denotes the Hadamard product. These masked updates are subsequently aggregated through $\Delta \mathcal{W}^t = \sum_{k=1}^K \lambda_k^t \Delta \mathbf{w}_k^t$, where $\lambda_k^t$ denotes the weighting coefficients derived from Eq. (11).

**Consensus Parameter Rescaling**. To enhance directional consensus in global updates, we amplify parameters exhibiting cross-client agreement through a rescale process. First, we construct the client update matrix $\Delta \mathcal{W}^t = [\Delta w_1^t, \ldots, \Delta w_K^t]^\top \in \mathbb{R}^{K \times G}$ and perform normalization:

$$\widehat{\Delta w}_k^t = \frac{\Delta \mathbf{w}_k^t}{\|\Delta \mathbf{w}_k^t\|}, \quad \widetilde{\Delta w}_{k,i}^t = \frac{\widehat{\Delta w}_{k,i}^t}{\sum_{j=1}^G \widehat{\Delta w}_{k,j}^t}. \tag{6}$$

This applies row-wise and column-wise normalization to $\Delta \mathcal{W}^t$, enabling cross-client comparability of update directions while preserving their relative importance.

The consensus degree of parameter $i$ is quantified by its standard deviation $\sigma_i^t = \text{std}(\{\widetilde{\Delta w}_{k,i}^t\}_{k=1}^K)$ across clients. We then amplify the most consistent parameters (those with the lowest $\sigma_i$) corresponding to the bottom $\rho$-quantile, where $\rho^t = \frac{1}{K} \sum_{k=1}^K r_k^t$ is the average mask rate. The purpose of using $\rho^t$ to determine the amplification set is to rescale the aggregated gradients back to their original magnitude, which is shown to benefit the model's performance [60, 15].

$$\alpha_i^t = \begin{cases} 1 + \frac{m_d^t}{m_a^t}, & i \in \mathcal{S}_{\text{amp}}^t = \{i \mid \sigma_i^t \leq \text{sorted}(\sigma^t)[(1 - \rho^t)G]\}, \\ 1, & \text{otherwise,} \end{cases} \tag{7}$$

where $m_d^t$ and $m_a^t$ denote the average magnitudes of discarded and amplified parameters respectively. This rescaling ensures model outputs remain stable relative to the pre-discarding state. The final aggregated update becomes $\mathcal{W}^{t+1} = \mathcal{W}^t + \alpha^t \odot \Delta \mathcal{W}^t$.

### 3.3 Adaptive Weighted Aggregation

**Motivation.** Our analysis of federated learning reveals staged characteristics in collaborative training, as shown in 4. The two sides of the green line exhibit different behaviors, representing two phases:

**Phase I: Early Training.** The initial phase exhibits strong client alignment, where test accuracy improves rapidly through collective gradient coherence. However, while most clients converge to beneficial update directions, the clients in the SYN domain show lower cosine similarity with other clients. These divergent directions may skew the dominant direction, compromising the generalization

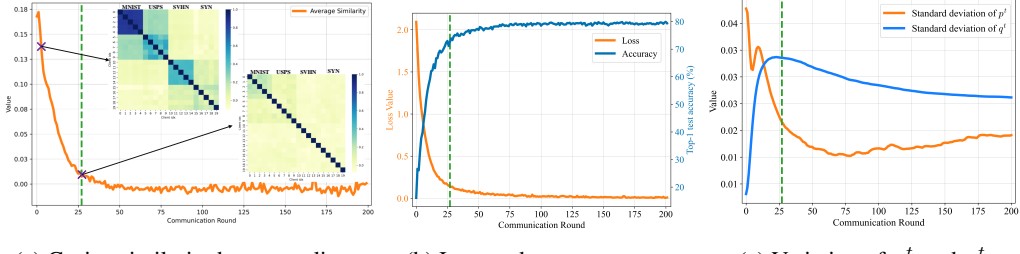

(a) Cosine similarity between clients    (b) Loss and accuracy curve    (c) Variation of $\sigma_p^t$ and $\sigma_q^t$

Figure 4: **Illustration of Training Dynamics**. The federated training has two phases: early phase (left of green line) and later phase (right of green line). The two phases exhibit distinctly different characteristics. Please refer to Sec. 3.3 for a detailed discussion.

ability of the global model. Our strategy therefore enforces directional consensus during this phase, selectively amplifying clients contributing to coherent, generalizable updates.

**Phase II: Later Training.** As training progresses, client updates evolve toward orthogonality (near-zero cosine similarity), signaling exhausted consensus-driven gains. Here, the very underperformers that posed risks in Phase I become valuable sources of exploratory gradient diversity. Their divergent directions now prevent premature convergence and enable fairer parameter distributions across heterogeneous clients. We correspondingly shift weighting priorities to elevate these previously marginalized contributors, transforming gradient conflicts into fairness-enhancing signals.

This phased paradigm fundamentally motivates our dual-term adaptive weighting framework. By dynamically rebalancing generalization and fairness priorities in response to emergent training signatures, we transcend the limitations of static aggregation schemes.

### 3.3.1 Consensus-Driven Generalization

The design of our generalization term stems from a fundamental intricate and interdependent relationship between gradient alignment patterns and collective learning progress dynamics. By analyzing the first-order Taylor expansion of the global loss reduction:

$$\Delta \mathcal{L}_{total}^t = \sum_{i=1}^{K} \Delta \mathcal{L}_i^t = \sum_{i=1}^{K} \left( \mathcal{L}_i^t \left( \mathbf{w}^t - \eta \mathbf{d}^t \right) - \mathcal{L}_i^t \left( \mathbf{w}^t \right) \right)$$

$$\approx \sum_{i=1}^{K} -\eta \cdot \langle \mathbf{g}_i^t, \mathbf{d}^t \rangle = \sum_{i=1}^{K} -\eta \cdot \langle \mathbf{g}_i^t, \sum_{j=1}^{K} \lambda_j \mathbf{g}_j^t \rangle \qquad (8)$$

$$= -\eta \cdot \sum_{i=1}^{K} \sum_{j=1}^{K} \lambda_i \langle \mathbf{g}_i^t, \mathbf{g}_j^t \rangle.$$

We establish that clients exhibiting higher gradient coherence (larger $\sum_j \langle \mathbf{g}_i^t, \mathbf{g}_j^t \rangle$) contribute more significantly to overall loss minimization. In practical training, $\mathbf{g}_j^t$ is equivalent to $\Delta \mathbf{w}_i^t$, and we assign more weight to those with a greater sum of dot products with other clients. This pairwise dot product formulation serves two purposes: it robustly captures directional consensus across skewed distributions through gradient similarity metrics, while simultaneously suppressing clients with divergent updates that could destabilize the global model. Additionally, we employ momentum updates [25], with the momentum coefficient decaying using a parameter $\gamma_p$, which helps constrain the oscillations that typically occur as the system approaches convergence.

$$\Delta p_m^t = (1 - \beta \gamma_p) \Delta p_k^{t-1} + \beta \gamma_p \frac{\sum_{i=1}^{K} \langle \Delta \mathbf{w}_i^t, \Delta \mathbf{w}_j^t \rangle}{\sum_{i,j} \langle \Delta \mathbf{w}_i^t, \Delta \mathbf{w}_j^t \rangle},$$

$$\gamma_p = \frac{\overline{\langle \Delta \mathbf{w}_i^t, \Delta \mathbf{w}_j^t \rangle}}{\langle \Delta \mathbf{w}_i^0, \Delta \mathbf{w}_j^0 \rangle}, \; p_k^t = p_k^{t-1} + \Delta p_k^t, \; p_k^t = \frac{p_k^t}{\sum_{j=1}^{K} p_j^t}, \qquad (9)$$

where $\overline{\langle \Delta \mathbf{w}_i^t, \Delta \mathbf{w}_j^t \rangle}$ represents the average value of the dot products between clients during epoch $t$.

### 3.3.2 Diversity-Enhanced Fairness

A natural approach to achieving fairness, as defined in (1), is to reweight the aggregation process by assigning higher weights to devices with poor performance. We use the loss as a proxy for performance and assign higher weights to clients with higher loss values, applying the same momentum

update to the fairness score as described previously.

$$\Delta q_k^t = (1 - \beta\gamma_q)\Delta q_k^{t-1} + \beta\gamma_q \frac{\mathcal{L}_i^t}{\sum_j^M \mathcal{L}_i^t}, \gamma_q = \frac{\overline{\mathcal{L}_i^t}}{\overline{\mathcal{L}_0^t}},$$

$$q_k^t = q_k^{t-1} + \Delta q_k^t, \quad q_k^t = \frac{q_k^t}{\sum_{j=1}^K q_j^t}. \tag{10}$$

To reduce the number of hyper-parameters, we adopt the same $\beta$ as in Eq. (9), and $\gamma_q$ serves the same role as $\gamma_p$. This mechanism aims to shift the weights in favor of disadvantaged clients, thereby achieving more uniform performance while also accounting for domain diversity.

### 3.3.3 Phase-Adaptive Weight Synthesis

Our phased analysis reveals a critical duality in federated optimization objectives: the alignment of updates and the disparity in loss exhibit opposing dominance patterns across different training phases. To empirically validate this phase-dependent dichotomy, we quantify the client-wise standard deviations of generalization ($\sigma_p^t$) and fairness ($\sigma_q^t$) weights throughout training. As shown in Fig. 4c, $\sigma_p^t$ dominates in the early rounds, while $\sigma_q^t$ surpasses it in later stages.

- **Early Phase**: $\sigma_p^t > \sigma_q^t$, high gradient alignment variability among clientsleads to prioritizing consensus-driven updates to maximize collective progress.
- **Later Phase**: $\sigma_q^t > \sigma_p^t$, the later-phase exhibits diminishing gradient coherence but increasing loss disparity, necessitating interventions that emphasize fairness.

This systematic inversion motivates our adaptive weighting mechanism, which dynamically rebalances the two objectives based on their relative variability:

$$\lambda_i = \frac{\sigma_p^t}{\sigma_p^t + \sigma_q^t}p_i^t + \frac{\sigma_q^t}{\sigma_p^t + \sigma_q^t}q_i^t. \tag{11}$$

This variance-regulated synthesis automatically shifts emphasis between consensus-seeking and disparity-reduction modes throughout the training dynamics.

### 3.4 Discussion and Limitation

**Comparison with Analogous Methods**. AFL [39], q-FFL [35], and FedCE [22] prioritize weak clients using single metrics (*e.g.* loss/accuracy), but simply increasing their weights is not appropriate. In contrast, our method first suppresses then amplifies updates for straggling clients, aligning with natural learning dynamics. This strategy maintains update consistency without sacrificing domain diversity. While prior works [44, 43] like FedLF address update conflicts via gradient projection, trivial parameters still disrupt global optimization. Our parameter adjustment selectively discards non-critical updates to preserve essential ones, enhancing both generalization and fairness.

**Discussion on Parameter Adjustment**. Our method addresses update conflicts in large-scale FL through selective parameter pruning. This strategy enables performance gains for conflicting clients with minimal self-degradation, achieving collective enhancement. Concurrent gradient rescaling preserves original gradient norm magnitudes while amplifying critical parameters, effectively reducing parameter space conflicts and stabilizing multi-client collaboration.

**Limitations**. Our method employs parameter adjustment and adaptive weighting to adjust model aggregation. However, setting the hyper-parameter $c$ to excessively high values may cause instability, exhibits sensitivity to selection. Our approach requires all participating clients to maintain identical network architecture specifications, which may limit the broader applicability of this method.

## 4 Experiments

We perform experiments on image classification tasks in various single-domain and cross-domain scenarios to validate the superiority of our framework FedPW.

### 4.1 Experiment Setup

**Datasets**. Following [18, 21, 43], we evaluate our method on single-domain datasets Fashion-Mnist [54], Cifar10 [28], Cifar100, and cross-domain datasets Digits [29] and Office-Caltech [11].

Table 1: Comparison of Average Accuracy (and Standard Deviation) with baselines in single-domain scenarios.

| Methods | FMNIST | | | CIFAR-10 | | | CIFAR-100 | | |
|---|---|---|---|---|---|---|---|---|---|
| | Dir(0.1) | Pat-1 | Pat-2 | Dir(0.1) | Pat-1 | Pat-2 | Dir(0.1) | Pat-1 | Pat-2 |
| FedAvg | 87.0(11.5) | 82.3(14.2) | 82.9(11.4) | 68.1(15.0) | 56.2(20.4) | 68.7(19.3) | 37.1(7.5) | 20.5(15.7) | 21.9(15.3) |
| q-FFL | 86.1(9.9) | 84.0(13.5) | 79.3(9.6) | 67.8(14.1) | 58.6(17.5) | 68.8(18.1) | 37.9(6.8) | 17.6(15.1) | 23.1(13.9) |
| AFL | 87.0(9.5) | 82.2(17.2) | 83.7(11.8) | 64.3(13.8) | 56.5(14.4) | 65.6(14.0) | 39.1(7.0) | 17.5(16.6) | 26.7(14.6) |
| Ditto | 79.9(10.6) | 75.5(21.4) | 77.8(10.2) | 57.9(13.1) | 45.7(11.3) | 52.6(14.2) | 27.9(7.4) | 18.1(20.1) | 11.2(11.4) |
| FedProx | 81.9(10.0) | 84.1(11.9) | 84.3(8.8) | 53.4(13.4) | 57.3(11.9) | 56.8(12.2) | 18.7(5.9) | 20.3(14.7) | 19.1(12.0) |
| FedFV | 82.8(9.4) | 88.5(11.7) | 81.6(13.4) | 68.1(13.3) | 55.1(18.3) | 69.4(18.9) | 37.3(6.9) | 18.9(14.5) | 21.9(13.7) |
| FedCKA | 89.0(10.1) | 79.3(18.8) | 87.4(10.9) | 67.3(14.3) | 56.4(20.4) | 69.3(14.4) | 37.1(7.0) | 19.6(15.7) | 21.3(14.4) |
| FedSAC | 84.1(13.1) | 74.5(23.6) | 83.4(18.1) | 58.7(14.3) | 44.8(15.3) | 55.8(15.3) | 33.4(6.1) | 25.4(14.3) | 7.5(3.1) |
| FedGCR | 85.7(10.0) | 83.2(13.5) | 84.0(11.3) | 66.6(13.8) | 61.5(19.1) | 65.5(16.9) | 38.6(6.7) | 20.6(14.5) | 19.9(15.6) |
| FedHeal | 85.7(11.3) | 85.8(13.9) | 84.0(11.6) | 71.0(14.4) | 58.7(22.4) | 66.8(14.1) | 38.7(6.8) | 19.6(14.9) | 22.8(14.3) |
| FedAA | 86.8(7.7) | 85.7(8.8) | 87.9(7.0) | 73.8(12.7) | 73.2(10.4) | 71.4(11.3) | 38.1(7.0) | 27.4(13.2) | 34.2(12.7) |
| **FedPW** | **88.2(7.1)** | **89.4(8.0)** | **90.2(6.6)** | **75.3(11.1)** | **75.1(10.3)** | **76.3(9.9)** | **41.2(6.6)** | **35.2(14.2)** | **38.4(12.6)** |

Table 2: Cross-domain comparison of Average Accuracy (**AVG**) and Standard Deviation (**STD**) with baseline.

| Methods | Digits | | | | | | Office-Caltech | | | | | |
|---|---|---|---|---|---|---|---|---|---|---|---|---|
| | MNIST | USPS | SVHN | SYN | AVG ↑ | STD ↓ | Amazon | DSLR | Caltech | Webcam | AVG ↑ | STD ↓ |
| FedAvg | 93.23 | 91.01 | 79.13 | 40.02 | 75.85 | 24.67 | 72.36 | 56.93 | 59.19 | 45.92 | 58.60 | 10.85 |
| +AFL | 93.73 | 93.42 | 75.42 | 44.25 | 76.71 $_{\uparrow 0.86}$ | 23.27 $_{\downarrow 1.40}$ | 64.34 | 65.65 | 57.21 | 47.52 | 58.68 $_{\uparrow 0.08}$ | 8.31 $_{\downarrow 2.54}$ |
| +q-FFL | 93.89 | 90.63 | 77.93 | 44.68 | 76.78 $_{\uparrow 0.93}$ | 22.48 $_{\downarrow 2.19}$ | 59.41 | 64.75 | 52.60 | 51.33 | 57.02 $_{\downarrow 1.58}$ | 6.26 $_{\downarrow 4.59}$ |
| +FedHEAL | 93.12 | 94.12 | 79.13 | 46.36 | 78.18 $_{\uparrow 2.33}$ | 22.29 $_{\downarrow 2.38}$ | 67.49 | 66.81 | 59.82 | 54.83 | 62.24 $_{\uparrow 3.64}$ | 6.03 $_{\downarrow 4.82}$ |
| **+FedPW** | 94.28 | 93.62 | 80.76 | 49.43 | **79.52** $_{\uparrow 3.67}$ | **21.00** $_{\downarrow 3.67}$ | 68.64 | 65.95 | 59.76 | 58.62 | **63.24** $_{\uparrow 4.64}$ | **4.83** $_{\downarrow 6.02}$ |
| FedProx | 93.64 | 91.14 | 80.53 | 41.93 | 76.81 | 23.94 | 70.31 | 57.83 | 59.86 | 44.79 | 58.20 | 10.48 |
| +AFL | 93.72 | 95.23 | 75.44 | 43.15 | 76.89 $_{\uparrow 0.08}$ | 24.22 $_{\uparrow 0.28}$ | 67.18 | 63.52 | 59.65 | 53.08 | 60.86 $_{\uparrow 2.66}$ | 6.03 $_{\downarrow 4.45}$ |
| +q-FFL | 94.05 | 93.49 | 75.73 | 44.36 | 76.91 $_{\uparrow 0.10}$ | 23.31 $_{\downarrow 0.63}$ | 62.27 | 73.62 | 54.39 | 55.46 | 61.44 $_{\uparrow 3.24}$ | 8.84 $_{\uparrow 1.64}$ |
| +FedHEAL | 92.15 | 93.58 | 78.89 | 44.61 | 77.31 $_{\uparrow 0.50}$ | 22.78 $_{\downarrow 1.16}$ | 66.17 | 72.65 | 58.09 | 56.95 | 63.46 $_{\uparrow 5.26}$ | 7.37 $_{\downarrow 3.11}$ |
| **+FedPW** | 94.33 | 92.46 | 80.19 | 48.62 | **78.90** $_{\uparrow 2.09}$ | **21.14** $_{\downarrow 2.80}$ | 68.40 | 70.67 | 59.86 | 58.94 | **64.47** $_{\uparrow 6.27}$ | **5.94** $_{\downarrow 4.54}$ |
| MOON | 92.65 | 92.81 | 80.51 | 39.63 | 76.40 | 25.18 | 73.01 | 60.29 | 59.66 | 47.54 | 60.13 | 10.40 |
| +AFL | 93.14 | 95.12 | 74.68 | 44.48 | 76.86 $_{\uparrow 0.46}$ | 23.46 $_{\downarrow 1.72}$ | 66.70 | 68.20 | 61.54 | 54.50 | 62.74 $_{\uparrow 2.61}$ | 6.18 $_{\downarrow 4.22}$ |
| +q-FFL | 92.31 | 94.51 | 75.98 | 43.67 | 76.62 $_{\uparrow 0.22}$ | 23.47 $_{\downarrow 1.71}$ | 64.90 | 65.85 | 53.88 | 58.93 | 60.89 $_{\uparrow 0.76}$ | 5.59 $_{\downarrow 4.81}$ |
| +FedHEAL | 93.23 | 94.31 | 80.81 | 45.12 | 78.37 $_{\uparrow 1.97}$ | 23.00 $_{\downarrow 2.18}$ | 68.16 | 64.95 | 59.17 | 59.51 | 62.95 $_{\uparrow 2.82}$ | 4.37 $_{\downarrow 6.03}$ |
| **+FedPW** | 93.71 | 94.61 | 81.53 | 48.27 | **79.53** $_{\uparrow 3.13}$ | **21.68** $_{\downarrow 3.50}$ | 66.73 | 65.84 | 59.47 | 60.34 | **63.10** $_{\uparrow 2.97}$ | **3.72** $_{\downarrow 6.68}$ |
| FedDyn | 94.15 | 94.82 | 80.29 | 40.48 | 77.44 | 25.53 | 70.11 | 61.56 | 59.78 | 48.15 | 59.90 | 9.04 |
| +AFL | 94.37 | 96.14 | 70.95 | 41.28 | 75.69 $_{\downarrow 1.75}$ | 25.65 $_{\uparrow 0.12}$ | 70.84 | 57.86 | 60.57 | 50.99 | 60.06 $_{\uparrow 0.16}$ | 8.24 $_{\downarrow 0.80}$ |
| +q-FFL | 94.71 | 94.26 | 75.33 | 42.79 | 76.77 $_{\downarrow 0.67}$ | 24.39 $_{\downarrow 1.14}$ | 62.99 | 66.44 | 55.76 | 55.36 | 60.14 $_{\uparrow 0.24}$ | 5.47 $_{\downarrow 3.57}$ |
| +FedHEAL | 94.61 | 95.72 | 79.94 | 43.72 | 78.50 $_{\uparrow 1.06}$ | 24.27 $_{\downarrow 1.26}$ | 67.55 | 60.53 | 58.86 | 53.38 | 60.08 $_{\uparrow 0.18}$ | 5.84 $_{\downarrow 3.20}$ |
| **+FedPW** | 94.61 | 95.72 | 80.06 | 46.84 | **79.31** $_{\uparrow 1.87}$ | **22.79** $_{\downarrow 2.74}$ | 66.93 | 63.27 | 58.83 | 54.67 | **60.93** $_{\uparrow 1.03}$ | **5.33** $_{\downarrow 3.71}$ |
| *Independent Methods* | | | | | | | | | | | | |
| Ditto | 93.62 | 91.58 | 79.55 | 40.65 | 76.42 | 24.65 | 56.94 | 69.67 | 56.73 | 56.82 | 60.04 | 6.42 |
| FedFV | 94.92 | 94.70 | 76.77 | 40.83 | 76.81 | 25.45 | 61.83 | 71.72 | 54.97 | 58.92 | 61.96 | 7.15 |
| FedCKA | 91.53 | 94.72 | 78.82 | 46.48 | 78.25 | 22.48 | 67.24 | 64.03 | 59.76 | 49.10 | 60.03 | 7.91 |
| FedGCR | 93.14 | 95.12 | 78.26 | 44.48 | 77.75 | 23.42 | 66.70 | 68.20 | 62.54 | 54.50 | 62.99 | 6.14 |
| FedSAC | 92.16 | 93.75 | 78.61 | 41.67 | 76.55 | 24.22 | 52.39 | 51.68 | 55.37 | 44.16 | 50.90 | 4.77 |
| FedAA | 92.91 | 92.28 | 78.57 | 47.16 | 77.73 | 21.43 | 62.89 | 68.71 | 56.83 | 56.02 | 61.11 | 5.92 |
| **FedPW** | 94.28 | 93.62 | 80.76 | 49.43 | **79.52** | **21.00** | 68.64 | 65.95 | 59.76 | 58.62 | **63.24** | **4.83** |

**Data Heterogeneity.**. To simulate heterogeneous clients in FL, we consider three scenarios: (1) Dir($\alpha$): We simulate $m$ clients in Dirichlet heterogeneous partition. The smaller $\alpha$ is, the more imbalanced the local distribution is. (2) Pat-1: It constructs a difficult data-island scenario where each client only has data from one class. (3) Pat-2: We follow FedAvg to build pathological non-IID data where each client has data from two classes.

**Model**. For the single-domain scenario, we conduct experiments with CNN(two convolutional layers) [53]. For cross-domain scenarios, we use ResNet-10 [14] whose feature vector dimension is 512. Note that all methods use the same network architecture for fair comparison across different tasks.

**Counterparts**. We compare our method with FedAvg [38] and fairness-focused FL approaches: AFL [39] , q-FFL [35], and FedHEAL[3] (both integrable). For non-integrable frameworks like Ditto [33], FedFV [53], FedGCR[4], FedSAC[56] and FedAA[13], we perform full end-to-end benchmarking. This ensures comprehensive evaluation across all baseline categories.

**Implement Details**. Following [3, 43], in the single-domain setting, we employ 100 clients for 3,000 communication epochs, where all federated learning methods exhibit minimal or no accuracy

improvement beyond this point. Each epoch involves 10% client participation. We use the SGD optimizer with a learning rate of 0.1 and a batch size of 50. For the cross-domain setting, we allocate 20 clients per task and equal clients per domain, with clients randomly assigned to domains. The training runs for E = 200 communication epochs with T = 10 local updates per round. Each epoch involves all clients. The SGD uses a learning rate of 0.001, and momentum is 0.9. The batch sizes are 64 for Digits and 16 for Office-Caltech. We fix the random seed to ensure reproduction and conduct experiments on the NVIDIA 3090Ti. The hyperparameter settings are detailed in Sec. 4.3.

**Evaluation Metric**. Following [34, 38], Top-1 accuracy is adopted for model performance evaluation. For the single-domain setting, we use the standard deviation of accuracy across clients, while for the cross-domain setting, we use the standard deviation across domains for fair evaluation. We conduct experiments three times and utilize the accuracy of the last five epochs as the final performance.

## 4.2    Comparison to State-of-the-Arts

We benchmark FedPW against contemporary approaches addressing Performance Fairness in FL, with comprehensive results presented in Tab. 1 and Sec. 4.1. Our method establishes new state-of-the-art performance, achieving superior mean accuracy while maintaining the lowest standard deviation across several scenarios. The convergence analysis in Fig. 5 further demonstrates FedPW's accelerated training dynamics compared to existing methods and several key observations are summarized:

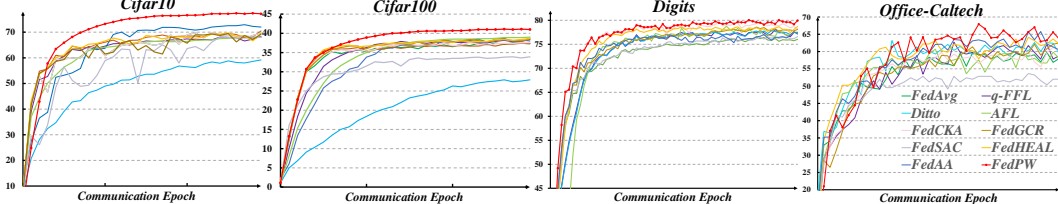

Figure 5: **Comparison of convergence of average accuracy** with counterparts. Please see details in Sec. 4.2.

❶ *FedPW achieves an optimal trade-off between overall performance and fairness.* Through the synergistic collaboration of the PA and AWA modules, FedPW simultaneously promotes fairness while improving the performance of the global model.

❷ *Existing fairness-oriented methods risk degrading the accuracy of global model or advantaged clients.* For instance, q-FFL and AFL exhibit performance degradation compared to FedAvg on Cifar-10, and similar phenomena occur with SVHN in the Digits benchmark.

❸ *FedPW safeguards performance for disadvantaged parties.* Underperforming domains like SYN in Digits and Webcam in Office-Caltech achieve improvements under FedPW's framework.

## 4.3    Diagnostic Experiments

**Compatibility Study**. To validate the compatibility of FedPW, we compared the results of several widely-adopted FL methods, FedAvg [38], FedProx [34], FedDyn [1], without and with FedPW. The results are shown in Sec. 4.1.

Table 3: **Ablation study** on multiple datasets. Please refer to Sec. 4.3 for detailed discussion.

| Setting | | FMNIST | | CIFAR-10 | | CIFAR-100 | | Digits | | Office-Caltech | |
|---|---|---|---|---|---|---|---|---|---|---|---|
| PA | AWA | AVG↑ | STD↓ | AVG↑ | STD↓ | AVG↑ | STD↓ | AVG↑ | STD↓ | AVG↑ | STD↓ |
| ✗ | ✗ | 87.03 | 11.21 | 68.13 | 15.02 | 37.12 | 7.48 | 75.85 | 24.67 | 58.60 | 10.85 |
| ✓ | ✗ | 88.05 | 10.34 | 72.88 | 13.94 | 38.64 | 7.81 | 77.85 | 23.62 | 60.65 | 9.92 |
| ✗ | ✓ | 87.34 | 8.62 | 72.12 | 12.84 | 40.01 | 7.27 | 78.58 | 22.08 | 62.39 | 5.05 |
| ✓ | ✓ | **88.22** | **7.13** | **75.28** | **11.10** | **41.22** | **6.57** | **79.52** | **21.00** | **63.24** | **4.83** |

**Ablation Study**. We conducted an ablation study to analyze the contributions of Parameter Adjustment (PA) and Adaptive Weighted Aggregation (AWA) components, as summarized in Tab. 3. Our findings indicate that each module positively contributes to performance, with optimal results achieved through their combination.

Table 4: AVG(STD) under different number of clients on Digits.

| Methods | Client scales | | |
|---|---|---|---|
| | 20 | 60 | 100 |
| FedAvg | 75.9(24.7) | 86.2(16.7) | 87.1(16.0) |
| FedPW | 79.5(21.0) | 88.9(9.8) | 91.3(7.8) |

**Hyper-parameter Analysis**. We systematically investigate the impact of two critical hyper-parameters: the client selection rate $c$ (Eq. (4)) and the momentum coefficient $\beta$ (Eq. (9)). Here, $\beta$ is solely used for momentum updates to mitigate drastic fluctuations during model training. Fig. 6 shows that $\beta$ has a limited impact on model performance, though the error bars indicate that increasing $\beta$ leads to more stable accuracy. The optimal values are set as defaults for subsequent experiments.

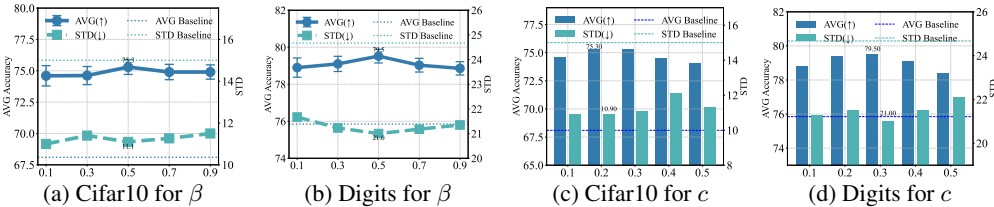

(a) Cifar10 for $\beta$      (b) Digits for $\beta$      (c) Cifar10 for $c$      (d) Digits for $c$

Figure 6: **Hyper-parameter study** with variant $\beta$ (Eq. (9)) and variant c (Eq. (4)). See details in Sec. 4.3.

## 5 Conclusion

In this paper, we explore the fairness challenges arising from domain skew in heterogeneous federated learning. We propose a simple yet effective federated learning algorithm, FedPW, to address two critical issues: Parameter Redundancy and Persistent Favoritism. Specifically, we utilize gradient information from model training to selectively discard and reinforce parameters. Furthermore, by leveraging training dynamics across epochs, our method achieves adaptive weighted aggregation. The effectiveness of FedPW has been extensively validated against several popular methods across various classification tasks. We hope that this work will serve as a foundation for future research.

## Acknowledgement

This work is supported by National Natural Science Foundation of China under Grant (62361166629, 623B2080, 62501428), the Major Project of Science and Technology Innovation of Hubei Province (2024BCA003, 2025BEA002), and the Innovative Research Group Project of Hubei Province under Grants 2024AFA017. The supercomputing system at the Supercomputing Center of Wuhan University supported the numerical calculations in this paper.

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

# A  Method Algorithm

---

**Algorithm 1:** FedPW

---

**Input:** Communication rounds $T$, local epochs $\mathcal{E}$, number of participants $K$, $k^{th}$ participant private data
$D_k$, private model $w_k$.
**Output:** The final global model $w^T$

---

**Server**: initialize the global model $w^0$
**for** $t = 0, 1, 2, ..., T-1$ **do**
    **Client**:
    **for** $k = 1, 2, ..., K$ **in parallel do**
        $w_k^t \leftarrow \mathcal{W}^t$
        **for** $e = 1, 2, ..., \mathcal{E}$ **do**
            $w_k^t \leftarrow w_k^t - \eta \nabla \mathbf{CE}(w_k^t, D_k)$
    $\Delta w_k^t \leftarrow w_k^t - \mathcal{W}^t$
    **Server**:
    $\Delta \mathcal{W}_k^t \leftarrow \mathbf{FedPW}(\Delta w^t)$
    **for** $i = 1, 2, \ldots, G$ **do**
        $\mathcal{W}_i^{t+1} = \mathcal{W}_i^t + \Delta \mathcal{W}_i^t$

**FedPW**$(\Delta w^t)$ : **for** $k = 1, 2, \ldots, K$ **do**
    `/* Adaptive Weighting */`
    $p_k^t, \Delta p_k^t, \leftarrow (p_k^{t-1}, \Delta p_k^{t-1}, \beta)$ in Eq. (9)
    $q_k^t, \Delta q_k^t \leftarrow (q_k^{t-1}, \Delta q_k^{t-1}, \beta)$ in Eq. (10)
    $\lambda_k^t \leftarrow (p_k^t, p_k^t)$ in Eq. (11)
    `/* Parameter Adjustment */`
    **for** $i = 1, 2, \ldots, G$ **do**
        $\mathcal{M}_{k,i}^t \leftarrow$ Eq. (5)
        $\Delta \mathbf{w}_{k,i}^t = \Delta w_{k,i}^t \cdot \mathcal{M}_{k,i}^t$
        $\alpha_i^t \leftarrow (\Delta w_{k,i}^t)$ in Eq. (7)
        $\Delta \mathcal{W}_i^t = \alpha_i^t \sum_{k=1}^K \lambda_k \Delta \mathbf{w}_{k,i}^t$

---

**return** $\Delta \mathcal{W}_i^t$

---

**Parameter Adjustment**. This module refines client updates through a two-stage process. In the first stage, redundant parameters are pruned by computing client-specific mask rates $r_k^t$ from inverse training losses and applying binary masks $\mathcal{M}_{k,i}^t$ in Eq. (5). Parameters below the adaptive threshold $\tau_k$ are removed, eliminating about $(1 - r_k^t) \times 100\%$ of the least significant parameters and reducing aggregation noise. In the second stage, consensus-based rescaling is applied using Eq. (7): masked updates are normalized, consistent parameters are identified via cross-client standard deviation, and emphasized by scaling factors $\alpha_i^t$ to preserve gradient magnitude while reinforcing aligned updates that support stable convergence.

**Adaptive Weighting**. This module balances generalization and fairness across training. In each round, generalization weights $p_k^t$ are computed via gradient alignment (Eq. (9)), while fairness weights $q_k^t$ are derived from training losses (Eq. (10)). The two components are then adaptively combined using the variance-based rule in Eq. (11), shifting from alignment-driven weighting in early rounds to fairness-oriented weighting later, guided by the relative variability of the two distributions.

# B  Details of Experiments

**Datasets**. Following [18, 21, 3, 43], we evaluate the efficacy of our method on single-domain datasets Fashion-Mnist, Cifar10, Cifar100, and cross-domain datasets Digits and Office-caltech.

- **Fasion-MNIST** [54] has 60k train and 10k test examples from 10 classes.
- **Cifar10** [28] contains $50k$, $10k$ images for training, validation. Each image is in $32 \times 32$ size from 10 different classes, *e.g.*, airplanes, cars, and birds.
- **Cifar100** [28] contains $50k$ and $10k$ images with $32 \times 32$ for 100 classes.
- **Digits** [29, 20, 41, 45] ] includes four domains: MNIST(M), USPS (U), SVHN (SV) and SYN (SY) with 10 cat- 427 egories (digit number from 0 to 9).

- **Office-Caltech** [11] consists four domains: Caltech (C), Amazon (A), Webcam (W) and DSLR (D), which is formed of ten overlapping classes between Office31 [46] and Caltech-256 [12].

**Hyper-parameter Study**  Our method involves only two hyperparameters: the mask ratio $c$ and the momentum coefficient $\beta$. The mask ratio $c$ governs the average masking level in the PA component, where smaller values make FedPW resemble FedAvg and reduce potential gains, while excessively large values may cause instability due to insufficient trainable parameters. The coefficient $\beta$ controls the smoothness of adaptive updates in the AWA component; extremely small values degrade it to a non-momentum variant, while overly large values may delay the system's responsiveness to training dynamics. The following tables present comprehensive hyperparameter evaluation results.

Table 5: **AVG(STD) under varying** $\beta$ across different datasets.

| $\beta$ | 0.1 | 0.3 | 0.5 | 0.7 | 0.9 |
|---|---|---|---|---|---|
| FMNIST | 85.1(8.3) | 87.7(7.2) | **88.2(7.1)** | 87.2(7.8) | 86.7(6.8) |
| CIFAR-10 | 74.5(10.9) | 74.6(11.4) | **75.3(11.1)** | 75.0(11.3) | 74.9(11.6) |
| CIFAR-100 | 39.6(6.8) | 41.2(7.0) | **41.2(6.6)** | 39.8(6.3) | 39.3(6.6) |
| Digits | 78.8(21.7) | 79.1(21.4) | **79.5(21.0)** | 78.5(21.1) | 77.9(21.4) |
| Office-Caltech | 62.7(6.3) | 61.9(4.8) | **63.2(4.6)** | 62.7(5.9) | 62.3(5.8) |

Table 6: **AVG(STD) under varying** $c$ across different datasets.

| $c$ | 0.1 | 0.2 | 0.3 | 0.4 | 0.5 |
|---|---|---|---|---|---|
| FMNIST | 86.1(7.1) | 87.6(7.4) | **88.2(7.1)** | 86.3(6.9) | 86.1(6.7) |
| CIFAR-10 | 74.6(10.9) | **75.3(10.9)** | 75.3(11.1) | 74.5(12.1) | 74.1(11.3) |
| CIFAR-100 | 40.2(6.8) | 41.0(6.6) | **41.2(6.6)** | 39.3(6.9) | 38.6(7.4) |
| Digits | 78.8(21.3) | 79.4(21.5) | **79.5(21.0)** | 79.1(21.5) | 78.4(22.1) |
| Office-Caltech | 62.1(5.1) | 62.3(5.1) | **63.2(4.6)** | 60.1(3.8) | 58.9(4.2) |

