# OpenReview forum: "Rethinking Fair Federated Learning from Parameter and Client View"
_NeurIPS.cc/2025/Conference — NeurIPS 2025 poster_

### Official Review · Reviewer_bBPA · 2025-06-13

**Clarity:** 3
**Significance:** 3
**Originality:** 3
**Rating:** 5
**Confidence:** 3

**Summary:**

The paper aims at solving a challenge  in federated learning： existing fair FLs often focus on improving weak clients during the training process. While this enhances weak clients, it can hinder global optimization, leading to lower overall performance and persistent unfairness.

To tackle this issue, the paper proposes a new fair FL framework, FedPW, with two main parts：(1) it introduces a Parameter Adjustment technique using masks and rescaling to discard redundant parameters, thus preserving important updates and reducing conflicts. (2) the author(s) propose  a dynamic aggregation function which  adaptively weights clients in training process varies across two phases.

The experiments show that FedPW achieves better global model accuracy and fairness across clients from different domains compared to multiple baselines.

**Questions:**

1. How is the redundancy value in the Fig. 2 (right) y-axis evaluated?

2. The standard deviations are significant in Table 2, and it seems not to represent an ideal fairness level, which means that the accuracy performance disparity among clients reaches up to 20%?

**Ethical Concerns:**

["NO or VERY MINOR ethics concerns only"]

**Final Justification:**

- The authors provided well-developed code in the discussion phase and clarified how their method establishes the necessary condition for Pareto stability.

- The provided equations and algorithmic flow make sense, and they have committed to adding further improvements in the revision.

- Compared to existing baselines, this paper advances the fairness of model performance distribution among clients, though without significant improvements. However, I like the perspective of this paper, from handling parameter redundancy, which is novel and interesting.

**Limitations:**

The adaptive  Weighting may have some practical limitations:

(1) whether accuracy and fairness can achieve Pareto stability, and

(2) whether the final level of fairness can be controlled, which can be guaranteed in other multi-objective gradient descent-based fair FLs.

**Paper Formatting Concerns:**

1. line 164: there is no $q_k^t$ in Eq. (9)， perhaps you are referring to Eq. (10).

2. There is significant incoherence between formulas. For example, to fully understand Eq. (5), one must refer to Eq. (11), which reduces the coherence of the reading.

3. It would be better to provide a clear algorithmic workflow that connects the formulas logically.

4. Clear definitions for $\sigma_p$ and $\sigma_q$ are lacking

**Quality:**

3

**Strengths And Weaknesses:**

1. This work proposes a Parameter Adjustment + Adaptive Weighting strategy. The former addresses weight conflicts  in federated learning under multi-objective optimization, while the latter provides a solution to ensure balanced performance across clients from different domains.
The inclusion of the Early Phase/Later Phase criterion is a valuable addition that provides insights into when fairness should be integrated into the training process. The proposed solution is logically structured, and the technical details are well-explained, making it a useful contribution to the FL community.

However, the adaptive  Weighting may have some practical limitations:

(1) whether accuracy and fairness can achieve Pareto stability, and

(2) whether the final level of fairness can be controlled, which can be guaranteed in other multi-objective gradient descent-based fair FLs.

2. Another strength is the well-designed experiment setup, with sufficient baselines and a variety of experimental scenarios. The effectiveness of the experiments may make this paper a good new fair FL baseline for federated learning with cross-domain clients.

The corresponding weakness is that the experiments is not completely provided in the code link.

---

> ### Author Rebuttal · Authors · 2025-07-31
>
> Dear Reviewer bBPA:
>
> Thank you very much for your thoughtful questions and concerns about our work. We have carefully considered each comment and provided responses.
>
> ### Weaknesses
>
> **W1 & L1: Pareto stability guarantee.**
>
> A1: Thank you for your insightful comment. Since our method is not designed from a multi-objective optimization perspective, we did not explicitly provide a Pareto optimality analysis in the main paper. However, our approach satisfies the necessary conditions for achieving Pareto optimality under certain assumptions. Specifically, the global descent direction in our method is a weighted sum of individual client gradients. In the early stages of training, these gradients tend to be aligned (i.e., having positive pairwise inner products), and in later stages, they become nearly orthogonal. As long as the weights remain positive and not excessively large, the resulting global direction maintains a positive inner product with each client’s gradient, thereby qualifying as a Pareto Descent Direction. This property enables FedPW to approximate the Pareto frontier in practice, promoting a balanced trade-off between accuracy and fairness.
>
> **W2: The provided experimental code is incomplete.**
>
> A2: We sincerely apologize for the earlier incomplete release of our experimental code. The repository has now been fully updated to include all components necessary for reproduction, including complete datasets scripts, training and evaluation pipelines, and baseline implementations. We welcome you to re-evaluate the codebase, and we appreciate your understanding and feedback.
>
> ### Questions
>
> **Q1: How is the redundancy value in the Fig. 2 (right) y-axis evaluated?**
>
> A3: We apologize for the confusion. The y-axis in Figure 2 (right, Page 4) does **not** represent the redundancy value itself. Each point corresponds to a particular experiment (as in Figure 2 left), where the x-axis denotes the redundancy ratio and the y-axis shows the resulting test accuracy. For instance, in the CIFAR-10 setting, when the redundancy ratio reaches around 0.85, the corresponding accuracy is approximately 70%.
>
> **Q2: Large std in Table 2 suggests a suboptimal fairness level.**
>
> A4: Thank you for your insights and concerns. As shown in Table 2 (Page 8), the performance varies noticeably across different domains. However, in multi-domain federated fairness settings, significant performance variation across domains is common due to extreme non-IID conditions. Our work focuses precisely on evaluating robustness under such challenging scenarios. Even state-of-the-art fairness methods struggle to achieve small standard deviations in this setting. Despite the seemingly large deviation values, our method achieves substantially lower disparity compared to SOTA baselines, highlighting its effectiveness and robustness in enforcing fairness under practical FL conditions.
>
> ### Paper Formatting Concerns
> **F1: Incorrect reference of Equation (10) as Equation (9).**
>
> A5: Thank you for pointing this out. We sincerely apologize for the oversight. There was indeed a mix-up between the references to Equations (9) and (10). We will correct this citation in the final version.
>
> **F2 & F3: Explanation of equations and algorithmic flow.**
>
> A6: We appreciate your valuable suggestion. Due to the coupling between the two components of our method, some equations are cross-referenced, which may have disrupted the overall flow of the algorithm. To clarify the core workflow, we provide a simplified pseudocode that highlights the essential steps. We will include the complete pseudocode in the appendix of the final version and revise the corresponding descriptions to improve clarity.
>
> > For each client $k = 1, ..., K$:
> > //*Adaptive Weighting*:
> >	$p_k^t, \Delta p_k^t \leftarrow (p_k^{t-1}, \Delta p_k^{t-1}, \beta)$ from Eq.(9)
> >	$q_k^t, \Delta q_k^t \leftarrow (q_k^{t-1}, \Delta q_k^{t-1}, \beta)$ from Eq.(10)
> >	$\lambda_k^t \leftarrow f(p_k^t, p_k^t)$ from Eq.(11)
> >//*Parameter Adjustment*:
> > For $i = 1, ..., G$:
> >	$\mathcal{M}\_{k,i}^t \leftarrow$ mask computed from Eq.(5)
> >	$\Delta \mathbf{w}\_{k,i}^t = \Delta w_{k,i}^t \cdot \mathcal{M}\_{k,i}^t$
> >	$\alpha_i^t \leftarrow$ scaling from Eq.(7)
> >	$\Delta \mathcal{W}\_i^t = \alpha_i^t \sum_{k=1}^K \lambda_k \Delta \mathbf{w}\_{k,i}^t$
>
> **F4:  Clear definitions for $\sigma_p$ and $\sigma_q$.**
>
> A7: Thank you for the feedback. Since the definitions of $\sigma_p$ and $\sigma_q$ are relatively straightforward, we only described them in words in the original manuscript. Specifically, $\sigma_p$ denotes the client-wise standard deviation of $p_k$ (defined in Eq. (9)), and $\sigma_q$ denotes the standard deviation of $q_k$ (defined in Eq. (10)). For completeness and clarity, we provide their explicit mathematical definitions below:
> $$
> \sigma_p = \sqrt{\frac{1}{N} \sum_{k=1}^N \left( p_k - \frac{1}{N} \sum_{i=1}^N p_i \right)^2}
> $$
> $$
> \sigma_q = \sqrt{\frac{1}{N} \sum_{k=1}^N \left( q_k - \frac{1}{N} \sum_{i=1}^N q_i \right)^2}
> $$

---

> ### Comment · Reviewer_bBPA · 2025-08-07
>
> Thank you for your clarification and promised updates. I have considered these in my final score.

---

> > ### Author Response · Authors · 2025-08-08
> >
> > Dear Reviewer bBPA,
> >
> > We greatly appreciate your thoughtful feedback and support for our work. Your detailed suggestions have played a crucial role in refining our study. We deeply value the effort and expertise you have dedicated to this review process, and we remain truly grateful for your guidance and support.
> >
> > Best regards,
> > Authors

---

### Official Review · Reviewer_ixbu · 2025-06-19

**Clarity:** 4
**Significance:** 3
**Originality:** 4
**Rating:** 5
**Confidence:** 4

**Summary:**

This paper focuses on performance fairness in federated learning. It addresses two key challenges: (1) parameter redundancy, which can lead to inefficient model updates, and (2)  the persistent protection of weak clients, which may act as outliers and reduce overall model quality. To tackle these issues, the authors propose FedPW, a dual-pronged approach: the paper introduces a parameter adjustment strategy to reduce redundancy and proposes an adaptive weighting mechanism to dynamically balance client contributions based on their reliability. The authors conducted extensive experiments, achieving higher average accuracy and lower performance disparity compared to baseline FL methods and fairness-oriented approaches in both single-domain and cross-domain scenarios.

**Questions:**

1) How does FedPW generate client-specific parameter masks? Could the authors clarify the criteria for selecting which parameters to mask for each client?

2) What defines the transition between training phases? Are these boundaries based on empirical observations or adaptive metrics?

3) Please specify the communication settings between the client and server (e.g., local training epoch), as these details are critical for reproducibility.

**Ethical Concerns:**

["NO or VERY MINOR ethics concerns only"]

**Final Justification:**

The authors have resolved all the issues I raised,  particularly regarding the ariance-based fairness metrics and the resource costs raised by Reviewer NSd9. I acknowledge their efforts and will upgrade my score for this submission.

However, the authors are recommended to incorporate two key revisions in the final version:

-  The hyperparameter sensitivity analysis;

-  The validation under more extreme FL conditions.

**Limitations:**

Yes

**Quality:**

3

**Strengths And Weaknesses:**

Strengths:

1. The figures are well-designed and enhance comprehension. The writing is clear and logically structured, making the paper easy to follow.

2. The paper addresses a fundamental and meaningful question: Can an FL algorithm achieve fairness without compromising performance? The authors provide two key insights to mitigate performance degradation in fairness-driven methods.

3. The identification of parameter redundancy and persistent protection issues is well-articulated and supported by explicit evidence.

4. The proposed method is novel and elegantly addresses the stated challenges.

5. The extensive experimental results demonstrate consistent improvements in both performance and fairness.

Weaknesses:
1. Definition 3.1 introduces performance fairness as the variance of client losses. However, the rationale for choosing this metric over established fairness measures (e.g., Gini coefficient, max-min gap, or Jain’s index) remains unclear. Please justify this choice and compare its implications with alternative metrics.

2. The experimental scope omits validation under extreme conditions, such as large-scale FL settings (>100 clients) or more realistic heterogeneity (e.g., Dirichlet allocation with α=0.01). These scenarios are critical for assessing scalability and robustness.

3. The approach requires all clients to share identical model architectures, which may limit generalization.

---

> ### Author Rebuttal · Authors · 2025-07-31
>
> Dear Reviewer ixbu:
>
> Thank you for your thorough review and insightful feedback. We sincerely appreciate your time and effort. We hope that the following responses effectively address your concerns.
>
> ### Weaknesses
>
> **W1: Additional details on variance-based fairness metrics.**
>
> A1: As shown in the following table, We additionally evaluate fairness using max-loss, minimum accuracy, the group gap between top-10% and bottom-10% clients, Gini index, and Jain’s fairness index. These results show that our method substantially improves the performance of the most disadvantaged clients across diverse fairness definitions.
>
> *Table: Fairness Scores under Different Fairness Metrics in Digits*
>
> | Method  | max-loss $\downarrow$ | min-acc $\uparrow$ | group gap $\downarrow$ | Gini index $\downarrow$ | Jain's index $\uparrow$ |
> | ------- | --------------------- | ------------------ | ---------------------- | ----------------------- | ----------------------- |
> | FedAvg  | 0.097                 | 39.39              | 47.23                  | 0.410                   | 0.936                   |
> | q-FFL   | 0.091                 | 42.66              | 46.57                  | 0.257                   | 0.942                   |
> | FedHeal | 0.094                 | 41.95              | 47.46                  | 0.213                   | 0.945                   |
> | FedPW   | **0.080**                 | **44.59**              | **44.13**                  | **0.180**                   | **0.951**                   |
>
> **W2: Validation under more extreme FL conditions.**
>
> A2: We acknowledge the importance of evaluating performance under more challenging federated settings. Our original submission already includes experiments with 100 clients and Dirichlet allocation at $\alpha$ = 0.1 (see Tables 3 and 4). To further validate the robustness of FedPW, we have added new experiments covering larger client scales (from 120 to 300 clients) and more severe data heterogeneity with $\alpha$ ranging from 0.01 to 0.09. The updated results confirm that FedPW consistently outperforms other methods across all these extreme conditions, demonstrating strong scalability and adaptability.
>
> *Table: Sensitive Analysis on Number of Clients in Digits*
>
> | Number of clients | 120        | 150        | 200        | 300        |
> | ----------------- | ---------- | ---------- | ---------- | ---------- |
> | FedAvg            | 88.3(15.3) | 87.4(15.5) | 88.8(15.1) | 88.0(15.8) |
> | FedPW             | **91.9(7.5)**  | **92.3(7.1)**  | **93.2(7.3)**  | **92.8(7.1)**  |
>
> *Table: Sensitive Analysis on Dirichlet Heterogeneity in CIFAR-10*
>
> | $\alpha$ | 0.01       | 0.03       | 0.05       | 0.07       | 0.09       |
> | -------- | ---------- | ---------- | ---------- | ---------- | ---------- |
> | FedAvg   | 78.2(13.1) | 82.3(10.2) | 82.1(10.7) | 84.7(10.9) | 86.4(11.7) |
> | FedPW    | **81.3(7.5)**  | 81.8(**7.8**)  | **86.0(7.1)**  | **86.7(7.1)**  | **88.23(7.2)** |
>
> **W3: Discussion on the limitations of assuming homogeneous model architecture.**
>
> A3: Using homogeneous models is a common and convenient setting for fairness-oriented methods. Most fairness-driven FL works adopt this setting, and we follow the same convention. Nonetheless, our approach is compatible with extensions to heterogeneous models. Specifically, our aggregation strategies do not require model alignment, and the pruning mechanism can be adapted to partially shared components, such as aligned layers. Therefore, our framework retains flexibility for broader applicability in non-iid or partially heterogeneous environments.
>
>
> ### Question
>
> **Q1: What is the criterion for generating parameter masks?**
>
> A4: FedPW constructs client-specific parameter masks based on the magnitude of local updates. For each client $k$, we sort the elements of the update vector $\Delta w_k^t$ by absolute value and compute a threshold $\tau_k$ based on the client-specific mask ratio $r_k^t$. All parameters below this threshold are marked as redundant using a binary mask $M_k^t$, as described in Eq. (5). The mask ratio $r_k^t$ is determined by the inverse of the client’s smoothed training loss, meaning that underperforming clients retain more updates to aid their progress.
>
> **Q2: How is the transition between the two training phases defined?**
>
> A5: The transition between the two training phases in FedPW is not determined by a fixed round count or manually defined schedule. Instead, it is guided by an adaptive, data-driven mechanism. The transition is soft and continuous, governed by the dynamics of training signals rather than a clear boundary, allowing the optimization focus to shift gradually in response to the evolving state of training.
>
> **Q3: What communication settings were used in the experiments?**
>
> A6: For single-domain experiments, we used 3,000 communication rounds with local epoch = 1 and batch size = 50. For cross-domain experiments, we used 200 communication rounds with local epoch = 10. The batch size is 64 for the Digits dataset and 16 for Office-Caltech. Detailed communication and training configurations are described in the Implementation Details section of the paper (Page 8).

---

> > ### Comment · Reviewer_ixbu · 2025-08-02
> >
> > Thanks for the rebuttal. After reading the rebuttal from the authors and the comments from other reviewers, I think the authors have addressed the concerns well. The applicability of FedPW across diverse fairness metrics and scenarios is well discussed, demonstrating the stability of the method. I will raise my score to accept. Look forward to seeing the camera ready version.

---

> ### Author Response · Authors · 2025-08-05
>
> Dear Reviewer ixbu,
>
> Thank you for your valuable comments and the time you spent reviewing our work. Your feedback has played an important role in helping us improve the paper, and we’re grateful for the opportunity to refine our work through this process.
>
> Best regards,
>
> Authors

---

### Official Review · Reviewer_Jzhw · 2025-06-20

**Clarity:** 3
**Significance:** 2
**Originality:** 3
**Rating:** 3
**Confidence:** 4

**Summary:**

This paper proposes FedPW, a novel federated learning framework aimed at improving both global performance and fairness across clients. To address parameter redundancy, it selectively discards insignificant local updates and amplifies consensus parameters that exhibit cross-client agreement. Additionally, it introduces an adaptive aggregation strategy that adjusts client weights dynamically based on the training phase: reinforcing update alignment in early training and emphasizing fairness in later stages. Extensive experiments on single- and cross-domain tasks demonstrate the effectiveness and compatibility of the proposed approach.

**Questions:**

1. The mask ratio $c$ and momentum coefficient $\beta$ appear to be critical hyperparameters, yet their impact is only briefly discussed. A more thorough analysis of how these parameters influence performance would be valuable.

2. Given that masking, normalization, and consensus computation require non-trivial operations, can the authors quantify the additional cost (e.g., time per round, memory usage)?
3. Due to the trade-off relationship between the number of communication rounds and local epochs in federated learning systems, so how does the number of **local epochs** (or steps) affect FedPW’s performance?
4. Some minor issues: Some methods in the related work section lack references, such as FedCurve and Ditto.

**Ethical Concerns:**

["NO or VERY MINOR ethics concerns only"]

**Final Justification:**

While I appreciate the clarifications, I remain unconvinced by the claim that the proposed method is compatible with heterogeneous model structures. The current formulation appears to assume homogeneous architectures across clients, and the authors have not provided sufficient technical details or empirical support to justify generalization to heterogeneous settings.

Moreover, I find the overall completeness of the work to be lacking. In particular, more extensive experimental analysis would be necessary to better support the claimed generality and effectiveness of the method.

Therefore, while the paper presents some interesting ideas, I still have reservations about its readiness for publication at this stage.

**Limitations:**

yes

**Quality:**

3

**Strengths And Weaknesses:**

Strengths：
1. This paper presents a technically solid approach with a thoughtfully designed method that tackles practical issues in federated learning. Extensive experiments consistently demonstrate the effectiveness of the proposed method.

2. This paper addresses an important and practical problem in federated learning: achieving fairness without compromising global performance.
3. This paper is well-written and well-organized.

Weaknesses:
1.  This method introduces several hyperparameters, but the sensitivity analysis is relatively limited.

2. This method relies on a common model architecture among clients, which could restrict its applicability in practical federated settings where model heterogeneity is present.
3. The overall framework introduces multiple components, making the system relatively complex and potentially difficult to implement or reproduce. It may hinder adoption in practical federated settings, especially where computational or engineering resources are limited.

---

> ### Author Rebuttal · Authors · 2025-07-31
>
> Dear Reviewer Jzhw:
>
> Thank you for the insightful comment. We appreciate your recognition of the innovative aspects of our motivation and the clarity of our writing. Thank you for your time and effort in reviewing our paper. We hope that our responses below will address your concerns and further affirm the quality of our research.
>
> ### Weakness
>
> **W1 & Q1: Additional hyperparameter sensitivity analysis.**
>
> A1: Our method involves only two hyperparameters: the mask ratio $c$ and the momentum coefficient $\beta$. The mask ratio $c$ governs the average masking level in the PA component, where smaller values make FedPW resemble FedAvg and reduce potential gains, while excessively large values may cause instability due to insufficient trainable parameters. The coefficient $\beta$ controls the smoothness of adaptive updates in the AWA component; extremely small values degrade it to a non-momentum variant, while overly large values may delay the system’s responsiveness to training dynamics. To further examine sensitivity, we supplemented results across all datasets. Experiments indicate that the method performs robustly across a range of hyperparameter settings, showing stable behavior without strong sensitivity.
>
> *Table: Average Accuracy(and Standard Deviation) Under Varying $\beta$*
>
> |                | 0.1        | 0.3        | 0.5        | 0.7        | 0.9        |
> | -------------- | ---------- | ---------- | ---------- | ---------- | ---------- |
> | FMINST         | 85.1(8.3)  | 87.7(7.2)  | 88.2(7.1)  | 87.2(7.8)  | 86.7(6.8)  |
> | CIFAR-10       | 74.5(10.9) | 74.6(11.4) | 75.3(11.1) | 75.0(11.6) | 74.9(11.6) |
> | CIFAR-100      | 39.6(6.8)  | 41.2(7.0)  | 41.2(6.6)  | 39.8(6.3)  | 39.3(6.6)  |
> | Digits         | 78.8(21.7) | 79.1(21.4) | 79.5(21.0) | 78.5(21.1) | 77.9(22.4) |
> | Office-Caltech | 62.7(6.3)  | 61.9(4.8)  | 63.2(4.6)  | 62.7(5.9)  | 62.3(5.8)  |
>
> *Table: Average Accuracy(and Standard Deviation) Under Varying $c$*
>
> |                | 0.1        | 0.2        | 0.3        | 0.4        | 0.5        |
> | -------------- | ---------- | ---------- | ---------- | ---------- | ---------- |
> | FMINST         | 86.1(7.1)  | 87.6(7.4)  | 88.2(7.1)  | 86.3(6.9)  | 86.1(6.7)  |
> | CIFAR-10       | 74.6(10.9) | 75.3(10.9) | 75.3(11.1) | 74.5(12.1) | 74.1(11.3) |
> | CIFAR-100      | 40.2(6.8)  | 41.0(6.6)  | 41.2(6.6)  | 39.3(6.9)  | 38.6(7.4)  |
> | Digits         | 78.8(21.3) | 79.4(21.5) | 79.5(21.0) | 79.1(21.5) | 78.4(22.1) |
> | Office-Caltech | 62.1(5.1)  | 62.3(5.1)  | 63.2(4.6)  | 60.1(3.8)  | 58.9(4.2)  |
>
> **W2: Discussion on the limitations of assuming homogeneous model architecture.**
>
> A2: Using homogeneous models is a common and convenient setting for fairness-oriented methods. Most fairness-driven FL works adopt this setting, and we follow the same convention. Nonetheless, our approach is compatible with extensions to heterogeneous models. Specifically, our aggregation strategies do not require model alignment, and the pruning mechanism can be adapted to partially shared components, such as aligned layers. Therefore, our framework retains flexibility for broader applicability in non-iid or partially heterogeneous environments.
>
> **W3 & Q2: Discussion on implementation difficulty and resource costs.**
>
> A3: Thank you for your insights and concerns. Our method introduces two components, both operating at the aggregation stage, which are not difficult to implement in practice. These steps are lightweight and can be incorporated as plugins without modifying the client-side logic or disrupting existing FL frameworks. We demonstrate its ease of integration via multiple plug-in experiments in Table 2 (Page 8).
>
> From a resource perspective, there is no extra client-side cost, either in computation or communication. The server-side overhead is also relatively small and acceptable. We measured the time and memory consumption on an NVIDIA 3090Ti, as shown in the following table. Additionally, we have supplemented a complexity analysis of our method, where the local epoch is denoted as $E$, batch size as $B$, number of parameters as $G$, number of clients as $K$, and dataset size as $D_k$. Although the complexity varies across these methods, their actual overhead in real-world scenarios is negligible and far smaller than the cost during training.
>
> *Table: Time (s) and Memory (GB) Costs on Digits Datasets*
>
> | Method  | Client Time | Server Time | Client Mem | Server Mem |
> | ------- | ----------- | ----------- | ---------- | ---------- |
> | FedAvg  | **3.58**        | **0.16**        | 0.042      | 0.022      |
> | q-FFL   | **3.58**       | **0.16**        | 0.042      | **0.021**      |
> | FedHeal | **3.58**        | 0.19        | 0.045      | 0.022      |
> | FedPW   | **3.58**        | 0.21        | **0.042**      | **0.021**      |
>
> _Table: Complexity Analysis._
>
> | Method  | Client Time Complexity     | Server Time Complexity                                  | Client Mem Complexity | Server Mem Complexity |
> | ------- | -------------------------- | ------------------------------------------------------- | --------------------- | --------------------- |
> | FedAvg  | $O(E \cdot B \cdot G)$     | $O(K \cdot G)$                                          | $O(D_k + G)$          | $O(K \cdot G)$        |
> | FedPW   | $O(E \cdot B \cdot G)$     | $O(K \cdot G + K \cdot c \cdot \log(G)  + K^2 \cdot G)$ | $O(D_k + G)$          | $O(K \cdot G)$        |
>
> ### Question
>
> **Q3: Impact of local training epoch.**
>
> A6: To evaluate how FedPW performs under different numbers of local training steps, we conducted experiments on the Digits cross-domain benchmark with local epochs varying from 5 to 20. The results, shown in the appended table, demonstrate that FedPW consistently outperforms baseline methods across all configurations. This suggests that our method maintains strong robustness to local training length.
>
> *Table: Sensitive Analysis on Local Epoch in Digits*
>
> | local epoch | 5          | 10         | 15         | 20         |
> | ----------- | ---------- | ---------- | ---------- | ---------- |
> | FedAvg      | 72.3(25.8) | 75.9(24.7) | 76.6(23.8) | 78.3(24.1) |
> | q-FFL       | 73.5(22.7) | 76.8(22.5) | 77.8(23.0) | 78.9(22.1) |
> | FedHeal     | 74.8(22.7) | 78.2(22.3) | 78.1(22.9) | 79.5(22.6) |
> | FedPW       | **75.6(21.9)** | **79.5(21.0)** | **79.0(21.1)** | **81.1(21.6)** |
>
>
> **Q4: Missing references for some related methods.**
>
> A7: We sincerely apologize for this oversight and thank the reviewer for the helpful reminder. The missing citations for FedCurve, Ditto, and other relevant works will be properly included in the final version of the paper.
>
> Thank you for your valuable feedback and hope that our rebuttal adequately addresses your concerns!

---

> > ### Comment · Reviewer_Jzhw · 2025-08-02
> >
> > Thank you for your response. I have also read the other reviews and your replies to them. After considering all the discussions, I have decided to keep my original score.

---

> ### Author Response · Authors · 2025-08-02
>
> Dear Reviewer Jzhw,
>
> Thank you again for your time and thoughtful review. If you have any remaining concerns, we would sincerely appreciate the opportunity to address them and hope you might reconsider your evaluation. We truly value your feedback and hope our work can meet your expectations.
>
> With sincere thanks,
>
> Authors

---

### Official Review · Reviewer_NSd9 · 2025-06-25

**Clarity:** 2
**Significance:** 2
**Originality:** 2
**Rating:** 4
**Confidence:** 4

**Summary:**

The paper aims to overcome the bias that FL based models face by proposing FedPW, a fairness-aware federated learning method that addresses two challenges: (1) parameter redundancy, where unimportant parameter updates interfere with meaningful gradients, and (2) persistent client protection, where prioritizing weak clients across all rounds leads to convergence degradation. FedPW performs parameter masking/resizing based on local update magnitudes and applies a phase-adaptive client weighting scheme based on gradient alignment and local loss. Empirical evaluations demonstrate improved performance-fairness trade-offs.

**Questions:**

1) How do you distinguish between truly redundant updates and temporally small ones due to gradient noise or slow convergence?

2) Can you derive any convergence bound for the masked and rescaled FedAvg variant? Under what assumptions on gradient noise does your procedure remain unbiased?

3) How sensitive is the scheduler to the batch size, number of clients, or variance scaling?

4) Are other fairness metrics applicable in this work such as max-loss, minimum accuracy, or worst-group accuracy across clients?

**Ethical Concerns:**

["NO or VERY MINOR ethics concerns only"]

**Final Justification:**

Thanks for answering my questions. I have raised my score keeping in mind that theoretical formalizations still need some work.

**Limitations:**

yes

**Quality:**

2

**Strengths And Weaknesses:**

[S1] The paper relies on a known observation that fairness mechanisms may themselves cause unfairness or performance degradation, particularly due to persistent overweighting of weak clients and noisy parameter updates. This is in line with the theory of importance sampling that while importance weighting impacts machine learning models early in training, its effect diminishes over successive training iterations.

[S2] The proposition of a two-phased training heuristic (early: consensus-driven; late: fairness-driven), instantiated using statistical indicators from client gradients and loss, is an interesting bias-mitigation approach.

[S3] The evaluation covers both single-domain and cross-domain FL with baselines and ablations.

[W1] The central idea of parameter redundancy appears to be both under-theorized and lacks any formalization; The redundancy criterion is based entirely on local update magnitude, assuming small updates are unimportant. Is this assumption valid? This needs clarity because of a few points: Small updates could represent delayed or convergent directions. Update magnitude is highly sensitive to batch stochasticity and learning rate. Parameters interacting in high-dimensional manifolds (e.g., batchnorm scales) often have noisy gradients regardless of their importance. For this, I have a question: How do you distinguish between truly redundant updates and temporally small ones due to gradient noise or slow convergence?

[W2] The masking and rescaling steps (Eqs. 5–7) are the mathematical core of the method, but they are treated entirely as black-box heuristics. There is no attempt to analyze: (1) How masking affects the optimization trajectory (e.g., bias, variance, directionality). For this, I have a question: Can you derive any convergence bound for the masked and rescaled FedAvg variant? Under what assumptions on gradient noise does your procedure remain unbiased?

[W3] The dual-phase aggregation strategy (Eq. 11) is intuitive but lacks robustness or theoretical validation. The scheduler uses the relative standard deviations $\sigma_p$ and $\sigma_q$ of gradient consensus and loss weights, respectively. This implicitly assumes: (1) gradient directions align early in training ($\sigma_p$ high), and (2) loss disparities dominate later ($\sigma_q$ high). However, in practice, these signals can be noisy, correlated, or reversed depending on batch stochasticity, task diversity, or optimizer state. For this, my question is: How sensitive is the scheduler to the batch size, number of clients, or variance scaling?

[W4] The paper defines performance fairness as minimizing the variance of client losses (Definition 3.1). While this is common in FL literature, it may have limitations: variance is symmetric as it penalizes both over- and underperformance. But fairness typically aims to improve the worst-off groups. For this, my question is: are other fairness metrics applicable in this work such as max-loss, minimum accuracy, or worst-group accuracy across clients?

[W5] Several parts of the paper suffer from imprecise explanations or inconsistencies. For instance, in Figure 2 (parameter pruning analysis) The x-axis is not defined. Does "mask rate" refer to percentage of parameters, magnitude threshold, or layer-wise pruning?

---

> ### Author Rebuttal · Authors · 2025-07-31
>
> Dear Reviewer NSd9:
>
> We sincerely appreciate your time and effort in reviewing our paper. We hope that our responses below will address your concerns and further affirm the quality of our research.
>
> ### Weaknesses & Questions
>
> **W1 & Q1: How to distinguish truly redundant parameter updates from temporarily small updates caused by noise or slow convergence?**
>
> A1: Thank you for your insights and concerns. Neural networks are known to contain subnetworks that are functionally equivalent to the full model, implying the presence of redundant parameters. In our work, we treat parameters with consistently small updates as redundant, since their contribution to loss reduction is minimal. This perspective has been widely supported in prior studies [1, 2]. To address the concern that some parameters may have small updates due to noise or slow convergence, we clarify the following:
>
> (1) **Temporarily small updates:** Some parameters may have small updates due to gradient noise, stochasticity, or slow convergence. These temporarily small magnitudes do not reflect true redundancy. Such parameters are not persistently suppressed or discarded, so occasional masking of them does not cause significant negative impact.
>
> (2) **Persistent redundancy:** Truly redundant parameters tend to have consistently small updates over the long term. Our masking strategy captures this pattern and consistently discards such parameters.
>
> (3) **Long-term mask:** Our masking strategy operates continuously over the long term. This long-term perspective ensures that only persistently inactive parameters are consistently masked, while temporarily small updates may be masked occasionally but are not systematically suppressed.
>
> This distinction between temporarily small updates and persistent redundancy allows us to identify and mask unimportant parameters in a robust manner.
>
> [1] Picking winning tickets before training by preserving gradient flow. In International Conference on Learning Representations (ICLR), 2021.
>
> [2] Gradient flow in sparse neural networks and how lottery tickets win. In Proceedings of the AAAI Conference on Artificial Intelligence, 2022.
>
>
> **W2 & Q2: Convergence bound and bias analysis.**
>
> A2: Our current masking strategy is not strictly unbiased, as it favors the preservation of high-magnitude parameters. However, the resulting bias remains limited. We decompose the global update as:
> $$
> w^{t+1} = w^t - \eta \cdot \left( \alpha^t \odot \sum_k \lambda_k \mathcal{M}_k g_k \right),
> $$
>
> and define the total bias as:
> $$
> b := \mathbb{E}[\alpha^t \odot \sum_k \lambda_k \mathcal{M}\_k g_k] - \sum_k \lambda_k g_k = b_{\text{mask}} + b_{\text{amp}},
> $$
> with bounds:
> $$
> \|b_{\text{mask}}\| \le \sqrt{c_{\max}} \cdot G, \quad
> \|b_{\text{amp}}\| \le \frac{m_d}{m_a} \cdot G.
> $$
> Assuming $F(w)$ is $L$-smooth and $\mu$-strongly convex, we obtain the convergence bound:
> $$
> \mathbb{E}[F(w^T)] - F(w^*) \le \mathcal{O}\left( \frac{1}{T} + \eta^2 G^2 \left( \sqrt{c_{\max}} + \frac{m_d}{m_a} \right)^2 \right).
> $$
> In typical settings, $\sqrt{c_{\max}} < 1$, $\frac{m_d}{m_a} \ll 1$, and $\eta G \ll 1$, implying that the bias term $b$ remains significantly smaller than $1/T$. This guarantees convergence to a bounded neighborhood with controlled directional bias.
>
>
> **W3 & Q3: Dual-phase scheduler robustness and sensitive justification.**
>
> A3: To assess the scheduler’s sensitivity, we focus on **two key aspects**:
> (1) **Stability**: whether the dual-phase phenomenon that the scheduler relies on is consistently observable and stable across settings, and
> (2) **Effectiveness**: whether the scheduler can consistently achieve the intended objectives in both phases under different configurations.
>
> For (1), we confirm the presence of the dual-phase behavior in all tested settings and report the proportion of training epochs in each phase, denoted as $E_{\text{Phase I}}$ and $E_{\text{Phase II}}$. In general, $E_{\text{Phase II}}$ accounts for a larger portion of the training, indicating that the scheduler tends to prioritize fairness for most of the time.
> For (2), we quantify how concentrated the scheduler’s selection is in each phase. Specifically, we report $K \cdot \lambda_{\text{top-25\%-p}}$ and $K \cdot \lambda_{\text{top-25\%-q}}$, which represent the average weight assigned to the top 25% of clients (in terms of $p_k^t$ and $q_k^t$, respectively), scaled by the total number of clients $K$. Clients with higher $p_k$ (or $q_k$) values are those more beneficial for generalization (or fairness), and they are expected to be favored in the corresponding phase. A stable value of $K \cdot \lambda_{\text{top-25\%}}$ greater than 1 indicates that the scheduler consistently prioritizes these top clients as intended.
>
> The following tables present our results. The first two rows present the accuracy and corresponding standard deviation (in parentheses) for our method and FedAvg. These results suggest that the scheduler is not overly sensitive to batch stochasticity or system heterogeneity, and its decision rule remains stable and reliable across different settings.
>
> *Table: Sensitive Analysis on Bacth Size in Digits*
>
> | | 4          | 16         | 64         | 256        |
> |-|-|-|-|-|
> | FedAvg                                | 88.7(11.9) | 86.7(14.9) | 75.9(24.7) | 69.8(25.5) |
> | FedPW                                 | 92.3(9.0)  | 89.4(7.3)  | 79.5(21.0) | 77.3(21.5) |
> | $E_{\text{Phase I}}$                  | 0.47       | 0.33       | 0.36       | 0.35       |
> | $E_{\text{Phase II}}$                 | 0.53       | 0.67       | 0.64       | 0.65       |
> | $K \cdot \lambda_{\text{top-25\%-p}}$ | 1.32       | 1.35       | 1.29       | 1.33       |
> | $K \cdot \lambda_{\text{top-25\%-q}}$ | 1.45       | 1.31       | 1.38       | 1.41       |
>
> *Table: Sensitive Analysis on Number of Clients in Digits*
>
> |                                       | 4          | 12         | 20         | 60         | 100        | 200        |
> |-| ---------- | ---------- | ---------- | ---------- | ---------- | ---------- |
> | FedAvg                                | 47.6(27.6) | 68.7(28.9) | 75.9(24.7) | 86.2(16.7) | 87.1(16.0) | 88.6(15.5) |
> | FedPW                                 | 56.3(25.4) | 74.9(24.6) | 79.5(21.0) | 88.9(9.8)  | 91.3(7.8)  | 93.4(7.1)  |
> | $E_{\text{Phase I}}$                  | 0.41       | 0.35       | 0.36       | 0.39       | 0.44       | 0.42       |
> | $E_{\text{Phase II}}$                 | 0.59       | 0.65       | 0.64       | 0.61       | 0.56       | 0.58       |
> | $K \cdot \lambda_{\text{top-25\%-p}}$ | 1.35       | 1.32       | 1.29       | 1.26       | 1.23       | 1.25       |
> | $K \cdot \lambda_{\text{top-25\%-q}}$ | 1.41       | 1.37       | 1.38       | 1.34       | 1.27       | 1.26       |
>
> *Table: Sensitive Analysis on Variance Scaling (Max Value of $\sigma_p$ and $\sigma_q$)*
>
> |                                       | FMNIST     | CIFAR-10   | CIFAR-100 | Digits     | Office-Caltech |
> | ------------------------------------- | ---------- | ---------- | --------- | ---------- | -------------- |
> | Max $\sigma_p$                        | 0.035      | 0.112      | 0.259     | 0.045      | 0.031          |
> | Max $\sigma_q$                        | 0.038      | 0.058      | 0.183     | 0.032      | 0.025          |
> | FedAvg                                | 87.0(11.5) | 68.1(15.0) | 37.1(7.5) | 75.9(24.7) | 58.6(10.9)     |
> | FedPW                                 | 88.2(7.1)  | 75.3(11.1) | 41.2(6.6) | 79.5(21.0) | 63.2(4.6)      |
> | $E_{\text{Phase I}}$                  | 0.29       | 0.41       | 0.38      | 0.36       | 0.52           |
> | $E_{\text{Phase II}}$                 | 0.71       | 0.59       | 0.62      | 0.64       | 0.48           |
> | $K \cdot \lambda_{\text{top-25\%-p}}$ | 1.25       | 1.35       | 1.53      | 1.29       | 1.41           |
> | $K \cdot \lambda_{\text{top-25\%-q}}$ | 1.21       | 1.29       | 1.48      | 1.38       | 1.22           |
>
>
>
> **W4 & Q4: Additional details on variance-based fairness metrics.**
>
> A4: As shown in the following table, We additionally evaluate fairness using max-loss, minimum accuracy, the group gap between top-10% and bottom-10% clients, Gini index, and Jain’s fairness index. These results show that our method substantially improves the performance of the most disadvantaged clients across diverse fairness definitions.
>
> *Table: Fairness Scores under Different Fairness Metrics*
>
> | Method  | max-loss $\downarrow$ | min-acc $\uparrow$ | group gap $\downarrow$ | Gini index $\downarrow$ | Jain's index $\uparrow$ |
> | ------- | --------------------- | ------------------ | ---------------------- | ----------------------- | ----------------------- |
> | FedAvg  | 0.097                 | 39.39              | 47.23                  | 0.410                   | 0.936                   |
> | q-FFL   | 0.091                 | 42.66              | 46.57                  | 0.257                   | 0.942                   |
> | FedHeal | 0.094                 | 41.95              | 47.46                  | 0.213                   | 0.945                   |
> | FedPW   |**0.080**|**44.59**| **44.13**                  | **0.180**                   | **0.951**                   |
>
> **W5: Further clarification on Figure 2 and the x-axis.**
>
> A5: We apologize for your confusion caused. The x-axis labeled "mask rate" indeed refers to the percentage of parameters (ranked by magnitude) that are pruned. This definition aligns with the meaning of $r_k^t$ in our paper, representing the percentage of parameters being discarded. The left plot illustrates how performance varies across different domains as the mask ratio increases, where the inflection point indicates the maximum tolerable pruning ratio. The right plot displays the corresponding mask ratios and accuracy levels at which these inflection points occur across different datasets.

---

### Note · Authors · 2025-08-12

Dear AC and Reviewers,

We sincerely thank the AC for the efforts and all reviewers for their valuable insights. We are pleased that our contributions were found meaningful, and we summarize our work and the discussion as follows.

This paper proposes **FedPW**, a FL framework that improves both global performance and fairness. We address two key issues in existing FL methods, parameter redundancy and persistent protection of weak clients, which cause conflicting updates and biased optimization. To resolve them, we introduce: (1) a parameter adjustment strategy to weaken redundant conflicts while emphasizing consensus parameters, and (2) a two-phase weighting strategy that aligns updates early and focuses on fairness later. Experiments show that FedPW achieves higher accuracy and smaller performance gaps than baseline and fairness-oriented methods.

**Strong points**
- Solid claim, clear motivation, and novelty, offering a useful contribution (Jzhw, ixbu, bBPA).
- Well-articulated identification of redundancy and persistent protection issues, supported by evidence (ixbu).
- Two-phase heuristic is an interesting bias-mitigation approach, revealing when fairness should be integrated (NSd9, bBPA).
- Broad evaluation showing superior accuracy and fairness (NSd9, Jzhw, ixbu, bBPA).

**Weak points and our responses**
- Sensitivity analysis: We provided evaluations across datasets, batch sizes, client numbers, variance scaling, Dirichlet heterogeneity, and hyperparameters.
- Fairness metrics & cost: We added results for five fairness metrics and included computation cost experiments and complexity analysis.
- Theoretical foundation: We added analysis on convergence and fairness properties.

During the discussion phase, no new issues were raised, and some reviewers explicitly confirmed that our clarifications resolved their concerns.

For the revision, we will:
- Include a more comprehensive sensitivity analysis.
- Provide results on additional fairness metrics along with computational cost and detailed theoretical discussion.
- Revise descriptions of our overall algorithm and include pseudocode for improved clarity.

Finally, we greatly appreciate that several reviewers recognize the potential of our work to inspire further exploration in the FL community. We would like to emphasize that our work provides the community with a novel and critical fairness perspective. We believe it is worth publishing to stimulate further discussion.

Best regards,
The Authors

---

### Decision · Program_Chairs · 2025-09-17

**Decision:**

Accept (poster)

**Comment:**

This paper proposes FedPW, a federated learning approach to improve average performance and fairness. FedPW is based on two strategies: using dynamic masks to discard redundant parameter updates and upweight critical ones; using dynamic aggregation strategies to weight clients based on update directions and performance variations. After discussion, reviewers are overall satisfied and believe the paper makes a great contribution in federated learning, especially the technical novelty that conforms to the intuition and common beliefs, the extensive experiment evaluations on both single-domain and cross-domain tasks, and high writing quality. Hence, we recommend acceptance. Congratulations! Please make sure to incorporate reviews in the camera-ready version, especially adding discussion to address concerns on hyperparameter sensitivity analysis, applicability/limitations on heterogenous model architectures, and theoretical results such as convergence bounds.